# Curing Necrotic Angiodermatitis with an Intact Fish Skin Graft in a Patient Living with Diabetes

**DOI:** 10.3390/medicina58020292

**Published:** 2022-02-15

**Authors:** Dured Dardari, Corinne Lequint, Anne Christine Jugnet, Tatiana Bénard, Marie Bouly, Alfred Penfornis

**Affiliations:** 1Centre Hopitalier Sud Francilien, Diabetology Department, 91100 Corbeil-Essonnes, France; corinne.lequint@chsf.fr (C.L.); annechristine.jugnet@chsf.fr (A.C.J.); tatiana.benard@chsf.fr (T.B.); marie.bouly@chsf.fr (M.B.); Alfred.penfornis@chsf.fr (A.P.); 2LBEPS, IRBA, Université Paris Saclay, 91025 Evry, France; 3Paris-Sud Medical School, Paris-Saclay University, 91100 Corbeil-Essonnes, France

**Keywords:** case report, necrotic angiodermatitis, fish skin graft, diabetes

## Abstract

*Background and Objectives*: We describe a case of necrotic angiodermatitis. *Materials and Methods*: We used an intact fish skin graft to treat a patient living with diabetes, which was complicated by end-stage renal failure and arterial hypertension. The entire therapeutic procedure was carried out in ambulatory care without requiring the hospitalization of the patient. *Results*: The patient experienced a marked reduction in pain and complete epithelization of the lesion after 10 weeks of treatment. *Conclusion*: Our experience presents a new therapeutic approach to necrotic angiodermatitis.

## 1. Introduction

Ulcers of the lower limbs are predominantly of vascular etiology due to venous or arterial insufficiency, although they may also be of neuropathic or mixed etiology [1]. Rarer causes exist and should be considered in the event of an atypical ulcer and/or one that does not heal despite adequate treatment. Necrotic angiodermatitis (NA) is nevertheless considered to be the fourth vascular cause of leg ulcers requiring specific care. NA was originally described by Fernando Martorell in 1945 in four patients aged 50 to 70 years, who were all obese and presented uncontrolled hypertension. More recent studies have shown that NA occurs in elderly subjects aged 72–74.5 years on average [2,3], with a slight predominance of women (61–69.8%) [2,3,4]. Patients almost always have hypertension (94.9–100%) [2,3,4,5] and frequently diabetes (39.7–58%) [2,3,4,5]. Otherwise, the fish skin graft appears to convert the wound from an inflammatory to a healing stage, potentially promoting a normal healing process, effectively allowing normal healing processes to occur [6]. The technology provides a natural structure that contains proteins and fats (including omega 3), allowing stem cells and cells to migrate in the fish skin graft where they create new dermal tissue to seal the wound [6].

## 2. Case Presentation

Our 76-year-old male patient had been living with type 2 diabetes since the age of 34 years. His diabetes was associated with arterial hypertension and dyslipidemia hyperuricemia. His diabetes was further complicated by the following: (i) sensitive neuropathy diagnosed by an abnormal response to the monofilament test; (ii) severe proliferative retinopathy managed by a complete retinal pan-photocoagulation; (iii) ischemic heart disease with a triple coronary bypass in 2012; and (iv) end-stage renal failure treated since 2018 with hemodialysis on arteriovenous fistula at the rate of three weekly sessions.

The patient consulted our diabetology department in France in April 2021 after the appearance of an ulcer on his right leg (Figure 1) accompanied by significant pain but without trauma or diffuse infection of the soft tissues.

He showed no hyperthermia. The patient had been treated by his family physician with clavulanic acid 125 mg + amoxicillin 1 g per day for 7 days for a suspected skin infection. However, the patient described intense permanent pain. The ulcer was initially reddish before turning necrotic and finally developing into a necrotic lesion surrounded by erythematous purpura that gradually spread. Based on the evolution of the lesion, its clinical appearance, and the patient’s profile, necrotic angiodermatitis was diagnosed. Analgesic therapy was prescribed with oxycodone hydrochloride 5 mg four times a day. For family reasons and so as not to change his dialysis center, the patient refused hospitalization in the dermatology department and was instead managed by the diabetology outpatient service. As alternative at-home treatment, the patient was offered an intact fish skin graft produced by Kerecis^®^ (Figure 2).

The secondary dressing was changed by the home nurse, while the fish skin graft was absorbable on the surface of the ulcer. Between April and June 2021, the patient received a total of 10 fish skin grafts. He received consultation in the outpatient clinic every 3 weeks. The appearance of the wound gradually improved with the complete cessation of analgesic therapy in the 5th week of treatment and the complete epithelization of the lesion after the application of the 10th fish skin graft (Figure 3). 

## 3. Diagnostics

At the first consultation, the patient had the following diagnostics: creatinine 675 μmol/L, C-reactive protein 30 mg/L, hemoglobin A1C 6.1%, and body mass index 21 kg/m^2^. Doppler ultrasound showed no hemodynamically significant abnormality or aneurysmal lesion of the abdominal aorta, iliac arterial axis, or femoropopliteal arterial axis. Proximal atherosclerosis was predominant in the subrenal abdominal aorta (moderately tight stenosis evaluated at just over 50% reduction in surface area measured by planimetry), the periosteal segment of the right primary iliac artery (very tight stenosis with 65% reduction in surface area), and the mid-left superficial femoral artery (moderately tight stenosis with 50% reduction in surface area). Arterial blood supply to the feet was also normal, as the foot and plantar arteries showed a normal flow with the presence of diffuse mediacalcosis on the distal axis.

The patient is currently being monitored in the healing unit of our diabetology department. The ulcer treated with the intact fish skin graft presents complete epithelization. The patient remains asymptomatic in terms of pain and without any recurrence.

## 4. Discussion

Chronic ulcers are a major public health issue with an annual cost (direct and indirect) estimated at USD 3 billion in the United Kingdom [7] and more than USD 12 billion in the United States [8]. Besides the financial consequences, they have a considerable impact on the quality of life of patients [9]. NA is the fourth cause of leg ulcers of vascular origin (after arterial, venous, and mixed etiologies) [10]. Despite being responsible for 15% of hospitalizations for ulcers in dermatology [11], this pathology is unrecognized and underdiagnosed. Additionally known as Martorell’s ulcer [2,3] or hypertensive ischemic leg ulcer [2,3], NA was first reported in 1945 by cardiologist Fernando Martorell. These different names may cause confusion and add to the mystery surrounding the pathology [12]. 

The management of NA has three objectives: pain management, detection and treatment of other cardiovascular risk factors, and application of skin grafts. Localized care aims to stop the extension of the ulcer, reduce pain, and promote the different phases of wound healing, namely debridement, budding, and epidermalization. Autolytic, mechanical, or surgical debridement is a prerequisite for the application of skin grafts, which have shown their effectiveness in terms of acceleration of scarring process [13] and pain relief [14]. This simple and inexpensive technique is the first-line treatment for this type of ulcer. One advantage of skin grafting is that it can be applied to elderly patients regardless of their comorbidities. Negative pressure therapy can also be offered to manage this chronic ulcer. 

Pain control is the mainstay to reduce the symptom burden. The use of level III analgesics is common. Analgesics for neuropathic pain, equimolar mixture of oxygen and nitrous oxide (EMONO), and local anesthetics are especially useful for improving patient comfort. High-potency topical corticosteroids (clobetasol propionate) applied to the purpuric lesion for 3 to 7 days have also been shown to have an analgesic effect [15]. Furthermore, the control of cardiovascular risk factors is an integral part of the management of NA with particular attention given to the regulation of arterial pressure. 

Cell therapy products (CTPs) have been slowly advancing in the treatment of nonhealing ulcers over the past 20 years. The first large studies on CTPs were published in 2005 for use on venous leg ulcers [16]. Subsequent studies [17,18,19,20] used pig small intestine and other materials, mainly decellularized membranes from mammals, as well as cellularized skin substitutes and freeze-dried amnionic membranes. In 2013, decellularized fish skin, delivered as sterilized freeze-dried material, was approved in Europe and the USA by the Food and Drug Administration. This material has the benefit of not being treated with antibiotics and virus-inactivating methods, thereby leaving the natural omega-3 fatty acids in place. Fish skin used for skin grafting is also a byproduct of the food industry. Fish-derived CTP is both ecologically sustainable and rich in naturally occurring soluble molecules and omega-3 fatty acids. Omega-3 fats have a multitude of positive actions, including an anti-inflammatory function, and to some extent, antibacterial properties. It is important to stress the efficacy of fish skin graft therapy on analgesic plan, as our patient completely discontinued antipain therapy at an early stage in the 5th week, which is not usually obtained with other treatments for NA. This consequently leads to a clear improvement in quality of life.

## 5. Conclusions

In our case, for the first time, we describe the curing of NA by the application of a fish skin graft. The use of this therapy seems to open a new avenue for the out-of-hospital management of necrotic angiodermatitis.

## Figures and Tables

**Figure 1 medicina-58-00292-f001:**
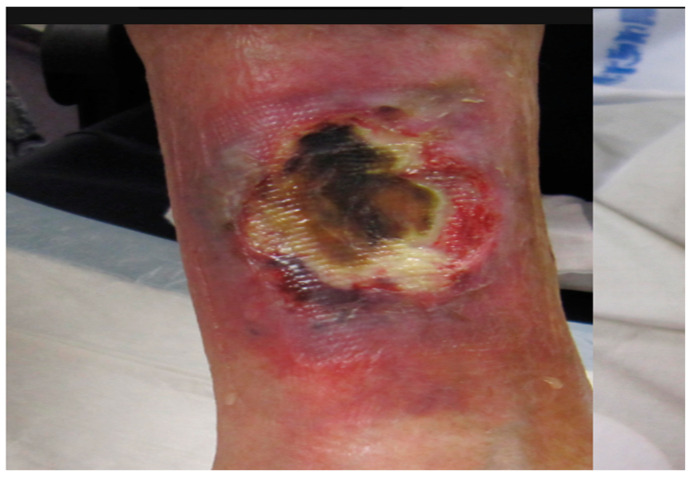
Necrotic angiodermatitis ulcer.

**Figure 2 medicina-58-00292-f002:**
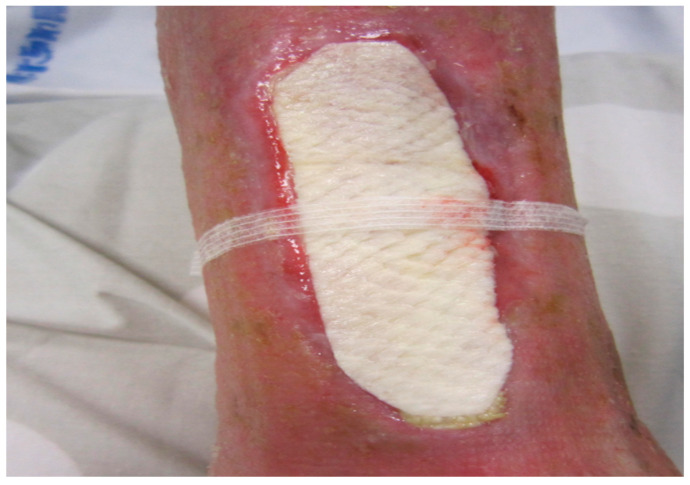
Ulcer and the application of intact fish skin graft.

**Figure 3 medicina-58-00292-f003:**
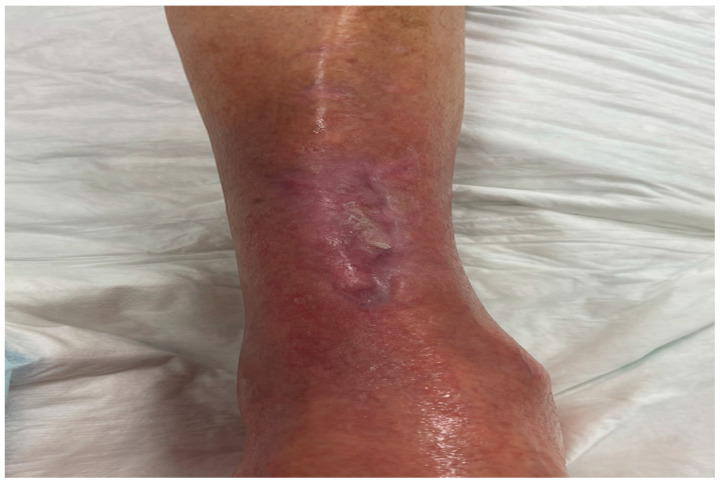
Complete epitalization of the lesion after application of the 10th fish skin.

## Data Availability

Data are available on request.

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
