# Peer review of "Curing Necrotic Angiodermatitis with an Intact Fish Skin Graft in a Patient Living with Diabetes"

_medicina, 2022, doi:10.3390/medicina58020292_

Round 1

Reviewer 1 Report

The paper has greatly improved. The part diagnostics could be shortenend. In the discussion, it could be mentioned in the paragraph about pain treatment, that the application of a skin graft or a skin substitute itself is often an important aspect of pain reduction

Author Response

The paper has greatly improved. The part diagnostics could be shortenend. In the discussion, it could be mentioned in the paragraph about pain treatment, that the application of a skin graft or a skin substitute itself is often an important aspect of pain reduction

Dardari answer: Many thanks for your comments: the diagnostics chapitre was shortened, a reference concerning pain reduction and skin grafting has been added.

Reviewer 2 Report

I think that the topic is of interest, but there are some points which have to be changed or improved:

The Introduction part is too short; I would recommend to write also some sentences about "fish skin" , because not everyone is expert in fish skin; I would recommend to write spmething about the product and its mode of actionen (based on the literature)! Some clinical picture of the wound prior and after treatment would be nice for the readers. Also the discussion part has to be more profound!!

Author Response

Many thanks for your comments, all your comments, all your comments have been taken into account and the text has been modified comments you have proposed it.